# Clinical evolution of ST11 carbapenem resistant and hypervirulent *Klebsiella pneumoniae*

Miaomiao Xie [1,5], Xuemei Yang[1,5], Qi Xu[1], Lianwei Ye[1], Kaichao Chen[1], Zhiwei Zheng[1], Ning Dong[1], Qiaoling Sun[2], Lingbin Shu[2], Danxia Gu[3], Edward Wai-Chi Chan[1,4], Rong Zhang [2✉] & Sheng Chen [1✉]

Carbapenem-resistant and hypervirulent *K. pneumoniae* (CR-HvKP) strains that have emerged recently have caused infections of extremely high mortality in various countries. In this study, we discovered a conjugative plasmid that encodes carbapenem resistance and hypervirulence in a clinical ST86 K2 CR-HvKP, namely 17ZR-91. The conjugative plasmid (p17ZR-91-Vir-KPC) was formed by fusion of a non-conjugative pLVPK-like plasmid and a conjugative $bla_{KPC-2}$-bearing plasmid and is present dynamically with two other non-fusion plasmids. Conjugation of p17ZR-91-Vir-KPC to other *K. pneumoniae* enabled them to rapidly express the carbapenem resistance and hypervirulence phenotypes. More importantly, genome analysis provided direct evidence that p17ZR-91-Vir-KPC could be directly transmitted from K2 CR-HvKP strain, 17ZR-91, to ST11 clinical *K. pneumoniae* strains to convert them into ST11 CR-HvKP strains, which explains the evolutionary mechanisms of recently emerged ST11 CR-HvKP strains.

[1] Department of Infectious Diseases and Public Health, Jockey Club College of Veterinary Medicine and Life Sciences, City University of Hong Kong, Kowloon, Hong Kong. [2] Department of Clinical Laboratory, Second Affiliated Hospital of Zhejiang University, School of Medicine, Zhejiang, Hangzhou, China. [3] Department of Clinical Laboratory, Zhejiang Provincial People's Hospital, Hangzhou, China. [4] State Key Lab of Chemical Biology and Drug Discovery, Department of Applied Biology and Chemical Technology, The Hong Kong Polytechnic University, Hung Hom, Kowloon, Hong Kong. [5] These authors contributed equally: Miaomiao Xie, Xuemei Yang. ✉email: zhang-rong@zju.edu.cn; shechen@cityu.edu.hk

K*lebsiella pneumoniae* is a major pathogen, which has become a critical causative agent of hospital infections in recent years[1,2]. Infections caused by *K. pneumoniae* are at present a major cause of morbidity and mortality[3], which was ascribed to the rapid evolution of *K. pneumoniae* in recent decades. Certain types of *K. pneumoniae* especially sequence type 23 (ST23) have developed into hypervirulent *K. pneumoniae* (HvKP). Initially collected from patients suffering severe liver abscess in late 1980s in Taiwan, HvKP was regarded to cause serious community-acquired infections among young and healthy individuals[4]. HvKP isolates can generate an enhanced level of capsule, which can be detected by a string test and positive result is iconic as a typical hypermucoviscous phenotype[5]. The hypervirulence of *K. pneumoniae* was ascribed to carriage of a virulence plasmid which harbors two capsular polysaccharides regulator genes (*rmpA* and *rmpA2*) and some siderophore determinants which encode the hypermucoviscous phenotype[6,7]. The interrelationship between the hypervirulence phenotype and virulence plasmid has already been demonstrated[8]. Majority of HvKP belonged to capsular serotype K1 such as ST23, and capsular serotype K2 including ST86, ST65, ST25, and ST375[9,10]. In addition, certain lineages of *K. pneumoniae* have also evolved into multidrug-resistant strains such as carbapenem-resistant *K. pneumoniae* (CRKP), which is also a clinically important pathogen that causes untreatable infections and is regarded as a pressing threat to human health[11]. CRKP remains an organism of relatively low virulence, whereas HvKP remains antibiotic-sensitive.

Recent studies have shown that traditional ST23 K1 HvKP strains have evolved further to become multidrug-resistant, especially carbapenem-resistant, through acquisition of plasmids that harbor various carbapnemase genes such as $bla_{KPC-2}$, $bla_{VIM}$, or $bla_{NDM-1}$[12–17]. Although being increasingly reported, this type of carbapenem-resistant HvKP (CR-HvKP) strains are still not very common, which is probably due to the low prevalence of HvKP strains in clinical settings. Originally low-level virulence CRKP strains such as ST11 CRKP have been shown to acquire the virulence plasmid from HvKP and evolved into hypervirulent CRKP strains that caused infections of high mortality in 2017[18]. Since this first report, CR-HvKP strains have been reported in various countries[19–22]. A particular concern is the emergence of virulence plasmid-bearing ST11 type CRKP strains, which belong to one of the most common clonal group 258 (CG258) type of CRKP reported worldwide. The emergence of ST11 CR-HvKP in Asia poses enormous challenges to infection control since this type of CR-HvKP has the potential to become as prevalent as ST11 CRKP. An increasing number of non-K1/K2 type CR-HvKP infections have been reported in various countries, which are

often associated with high mortality[19–22]. Mechanisms underlying the transmission of pLVPK-like virulence plasmid to non-K1/K2 type of *K. pneumoniae* are not clear as the virulence plasmid is not self-transmissible due to the lack of the *tra* gene locus for plasmid conjugation. This has probably limited the transmission of the virulence plasmid among non-K1/K2 type of *K. pneumoniae* strains previously. In this study, we characterized a conjugative fusion plasmid that encodes both carbapenem resistance and hypervirulence, uncovering a mechanism of evolution towards HvKP in non-K1/K2 strains. Transmission of such conjugative fusion plasmid among *K. pneumoniae* strains in hospitals could cause a rapid increase in the prevalence of fatal CR-HvKP infections, posing a grave threat to public health.

## Results

**Characteristics of carbapenem-resistant *K. pneumoniae* 17ZR-91**. *K. pneumoniae* 17ZR-91 exhibited resistance to all β-lactam antibiotics tested, but remained susceptibility to amikacin, ciprofloxacin, polymyxin B, and tigecycline (Table 1). Results of string test showed that *K. pneumoniae* 17ZR-91 generated $a > 5$-mm viscous string, indicating the hypermucoviscous phenotype of this strain[5]. Complete genome sequence of 17ZR-91 was successfully obtained. Strain 17ZR-91 was identified as belonging to ST86 (*gapA-infB-mdh-pgi-phoE-rpoB-tonB* allele number 9-4-2-1-1-1-27) and K2 serotype based on capsular typing by *wzi* allele[23,24]. Blast against the resistance and virulence genes database revealed that strain 17ZR-91 contained several virulence determinants, including the regulator of mucoid phenotype A2 (*rmpA2*) and mucoid phenotype A (*rmpA*), aerobactin (*iucABCD*, *iutA*), salmochelin (*iroBCDN*), yersiniabactin (*ybt9*; *ICEKp3*), type 3 fimbriae-encoding system (*mrkABCDFHIJ*), as well as a number of resistance genes including $bla_{KPC-2}$, $bla_{SHV-2}$, *oqxAB*, *mph*(A), *sul1*, *aadA2* and *dfrA12*. These data further confirmed that strain 17ZR-91 belonged to ST86 K2 HvKP that exhibited the carbapenem resistance phenotype.

**Identification of conjugative plasmid encoding carbapenem-resistance and hypervirulence in *K. pneumoniae* 17ZR-91**. Plasmids harbored by *K. pneumoniae* 17ZR-91 were investigated. S1-PFGE analysis illustrated that two plasmids were recoverable from strain 17ZR-91 (Fig. 1a). Both Illumina and Nanopore platforms were used to obtain the complete sequences of these two plasmids. Consistent with S1-PFGE results, two circular plasmids of 215,723 bp and 114,834 bp were obtained. The 215,723 bp plasmid contained an IncHI1 replicon, and comprised 234 predicted coding sequences with a GC content of 50.4%, which harbored virulence-related determinants (*rmpA2*, *rmpA*,

**Table 1 Characteristics of carbapenem-resistant *K. pneumoniae* 17ZR-91 and corresponding transconjugants.**

| Strain ID | Species | ST | MIC (µg mL⁻¹) | | | | | | | | | | | *rmpA2*, $bla_{KPC-2}$ | Conjugation efficiency |
|---|---|---|---|---|---|---|---|---|---|---|---|---|---|---|---|
| | | | IMP | ETP | MRP | CTX | CAZ | ATM | AMK | CIP | TIG | PB | TE | | |
| 17ZR-91 | *K. pneumoniae* | ST86 | 4 | 32 | 2 | 32 | 16 | 128 | 1 | 0.03 | 0.5 | 1 | >128 | + | NA |
| EC600 | *E. coli* | – | 0.25 | 0.015 | 0.03 | 0.06 | 0.5 | 0.12 | 0.5 | 0.12 | 0.5 | 0.5 | 2 | − | NA |
| EC600-TC | *E. coli* | – | 4 | 8 | 2 | 16 | 8 | 64 | 0.5 | 0.12 | 0.25 | 0.5 | 64 | + | 1.67E-08 |
| KP04-1 | *K. pneumoniae* | ST11 | 0.12 | 0.5 | 0.06 | >128 | >128 | >128 | 16 | >32 | 2 | 1 | 1 | − | NA |
| KP04-1-TCª | *K. pneumoniae* | ST11 | 16 | 16 | 8 | >128 | >128 | >128 | 16 | 32 | 0.5 | 1 | 128 | + | 4.58E-06 |
| KP04-1-TC1 | *K. pneumoniae* | ST11 | 16 | 16 | 8 | >128 | >128 | >128 | 16 | >32 | 0.5 | 1 | 128 | + | 2.43E-06 |
| KP04-1-TC2 | *K. pneumoniae* | ST11 | 16 | 16 | 8 | >128 | >128 | >128 | 16 | >32 | 0.5 | 1 | 128 | + | 2.43E-06 |

'NA' indicates that the test or analysis was not conducted on this strain.
KP04-1-TC1 and KP04-1-TC2 are two different transconjugant colonies from the conjugation experiments.
*CAZ* ceftazidime, *CTX* cefotaxime, *IMP* imipenem, *MRP* meropenem, *ETP* ertapenem, *AMK* amikacin, *CIP* ciprofloxacin, *PB* polymyxin B, *ATM* aztreonam, *TE* tellurite, *TIG* tigecycline.
ªTransconjugants obtained using *E. coli* EC600-TC as donor.

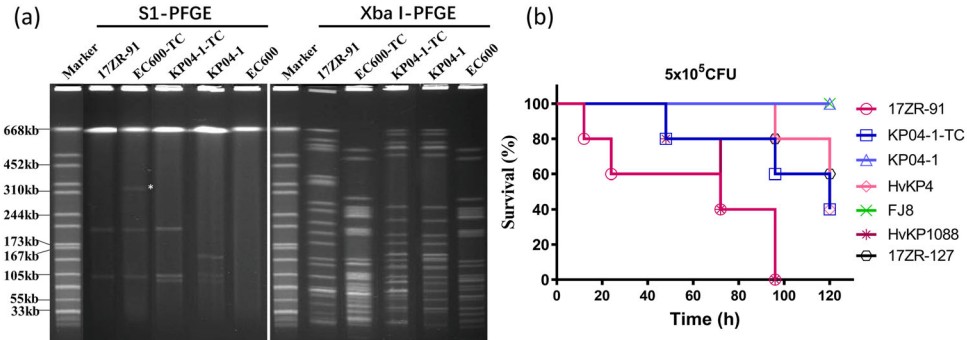

**Fig. 1 PFGE profiles and virulence discrepancy of K. pneumoniae 17ZR-91 and corresponding transconjugants. a** S1-PFGE and XbaI-PFGE profiles of *K. pneumoniae* 17ZR-91 and different transconjugants. Asterisk, conjugative fusion virulence plasmid. EC600-TC, transconjugant obtained by conjugation of plasmids from 17ZR-91 to *E. coli* EC600 and three plasmids were observed in this strain; KP04-1-TC, transconjugant obtained by conjugation of plasmids from EC600-TC to ST11 classical *K. pneumoniae* strain, KP04-1 and three plasmids were observed in this strain. **b** Virulence potential of *K. pneumoniae* isolates tested in a mouse infection assay. Survival of mice infected with 5*10$^5$ CFU of each *K. pneumoniae* at 120 h was shown (*n* = 10). Statistical analysis by the Log-rank (Mantel–Cox) test was conducted for indicated curves.

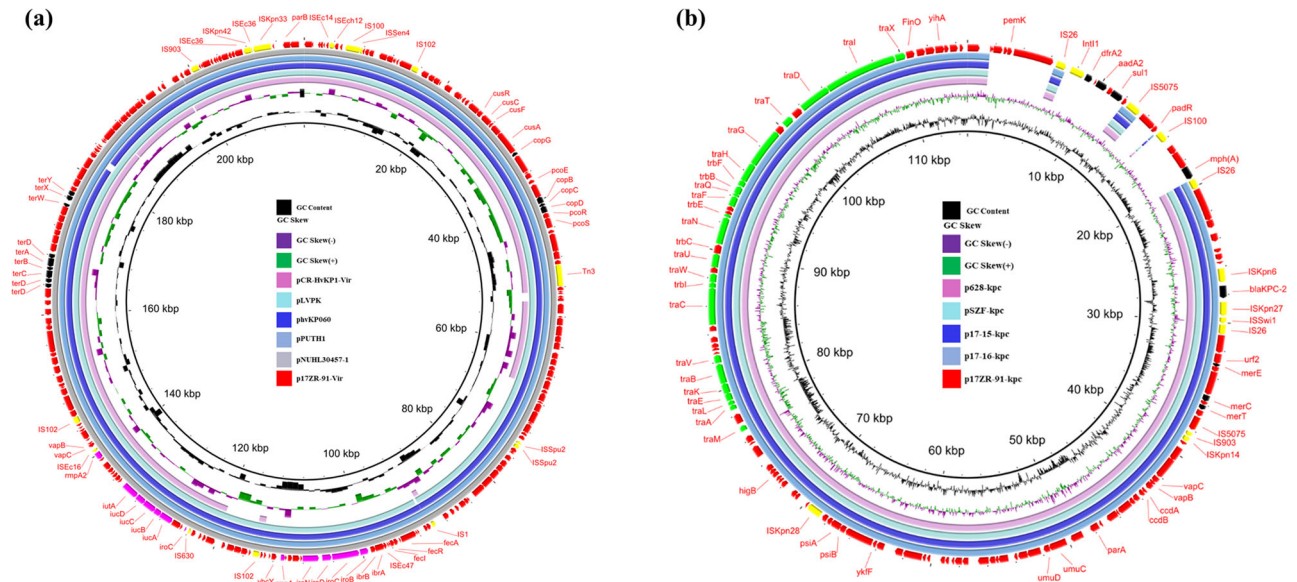

**Fig. 2 Circular alignment of virulence plasmid p17ZR-91-Vir and bla$_{KPC-2}$-bearing plasmid p17ZR-91-KPC recovered from K. pneumoniae 17ZR-91 with similar plasmids in the NCBI database, respectively. a** Alignment of virulence plasmid p17ZR-91-Vir with pLVPK (AY378100), phvKP060 (CP034776), pPUTH1(CP024708), and pNUHL30457-1(CP026587.1) using BRIG. **b** Alignment of *bla*$_{KPC-2}$-bearing plasmid p17ZR-91-KPC with p628-kpc(NC_032103), pSZF-KPC (MH917122), p17-15-KPC (MK183753), and p17-16-KPC(MK191023) using BRIG.

*iutAiucABCD* and *iroBCDN*) and genes encoding resistance to tellurium and copper (*terABCD* and *pcoBCD*). It was speculated to be a virulence plasmid and designated as p17ZR-91-Vir (Fig. 2a). BLAST results showed that it exhibited high level of homology (99.93% identity and 99% coverage) with the classical virulence plasmid pLVPK (AY378100) and other virulence plasmids reported previously, such as phvKP060 (CP034776), pPUTH1(CP024708) and pNUHL30457-1(CP026587) (Supplementary Fig. S1a). The 114,834 bp plasmid contained an IncFII replicon, and comprised 142 predicted coding sequences with a GC content of 54.6%. This plasmid, which carried an incomplete transposon △TnSF1 and several antibiotic resistance genes including *bla*$_{KPC-2}$, *mph*(A), *sul1*, *aadA2*, and *dfrA12* genes, was designated as p17ZR-91-KPC (Fig. 2b). BLAST results indicated that p17ZR-91-KPC displayed a high level of homology (99.95% identity and 86% coverage) with other *bla*$_{KPC-2}$-bearing plasmids such as p628-kpc(NC_032103), pSZF-KPC (MH917122), p17-15-KPC (MK183753), and p17-16-KPC(MK191023). The main

difference between p17ZR-91-KPC and other *bla*$_{KPC-2}$-bearing plasmids lies in the region which contains MDR-encoding mobile elements (Supplementary Fig. S1b). Conjugation experiments were performed, with results confirming that p17ZR-91-KPC was a conjugative plasmid with conjugation efficiency of $2.45 \times 10^{-5}$. This finding is consistent with the identification of the *tra* gene locus in this plasmid, which is responsible for plasmid conjugation.

In order to evaluate the conjugative potential of virulence plasmid p17ZR-91-Vir, conjugation experiment was conducted using *K. pneumoniae* 17ZR-91 as the donor and *E. coli* EC600 as recipient. Under selection by potassium tellurite, a virulence plasmid-bearing transconjugant EC600-TC was obtained. Antibiotic susceptibility of EC600-TC was performed, with results showing that EC600-TC exhibited not only resistance to potassium tellurite, a phenotype encoded by the virulence plasmid, but also resistance to various β-lactam antibiotics including carbapenems, suggesting that, in addition to the

virulence plasmid, EC600-TC also acquired a MDR plasmid (Table 1). PCR assays targeting the *rmpA2* gene and *bla*$_{KPC-2}$ gene were performed as described previously[18] on five randomly selected transconjugants. The results showed that all transconjugants harbored these two genes, confirming that they have obtained the virulence plasmid and the *bla*$_{KPC-2}$-bearing plasmid. The genetic identity of transconjugant EC600-TC was also verified to be identical to *E. coli* EC600 as they exhibited identical XbaI-PFGE profile (Fig. 1a). Interestingly, S1-PFGE analysis showed that two plasmids were harbored by strain 17ZR-91, yet transconjugant EC600-TC was found to carry three plasmids (recipient *E. coli* EC600 did not harbor any plasmids), with two being identical to the plasmids in donor and one plasmid of a larger size that might be formed via co-integration of the two smaller plasmids (Fig. 1a).

**Genetic features of conjugative fusion plasmid.** To investigate the genetic features of the three plasmids harbored by *E. coli* transconjugant EC600-TC, plasmid sequencing using the Illumina and Nanopore platforms was performed. Our data confirmed that this strain carried three plasmids. Two smaller plasmids of size 215,723 bp and 114,834 bp were identical to p17ZR-91-Vir and p17ZR-91-KPC, respectively. The large plasmid turned out to be a fusion plasmid formed by recombination of p17ZR-91-Vir and p17ZR-91-KPC (Fig. 3). Plasmid analysis confirmed that the fusion event occurred via interaction at a short homologous region, which was a 275 bp DNA region encoding a hypothetical protein. One copy each of this homologous region (HR) could be detected in p17ZR-91-Vir (HR1) and p17ZR-91-KPC (HR2) (Figs. 3a, b). It should be noted that HR1 and HR2 in these two plasmids were not identical but shared 82% homology (Supplementary Fig. S2). According to these results, we hypothesize that two non-fusion plasmids were aligned at HR1 and HR2 and underwent homologous recombination, resulting in the fusion plasmid p17ZR-91-Vir-KPC (Fig. 3c).

To further study the genetic basis of conjugation of two non-fusion plasmids in strain 17ZR-91 to *E. coli* EC600, we determined if the fusion plasmid p17ZR-91-Vir-KPC might also be present in 17ZR-91 and co-exist with two non-fusion plasmids in a dynamic manner which has previously been observable in other type of fusion plasmid[25]. We analyzed the Nanopore reads of plasmids from strain 17ZR-91 and found seven reads covering the fusion region linked by HR1 and HR2, confirming the presence of fusion plasmid p17ZR-91-Vir-KPC in strain 17ZR-91 (Supplementary Fig. S3). In addition, PCR assays targeting the fusion regions as described in Figure S4a were performed. Our data showed that strain 17ZR-91 was positive for PCR1 and PCR2, which were targeting at the two fusion regions in p17ZR-91-Vir-KPC, suggesting the presence of p17ZR-91-Vir-KPC in 17ZR-91 (Supplementary Fig. S4b). In addition, positive detection of PCR3 and PCR4 products suggested the presence of two non-fusion plasmids, a finding consistent with S1-PFGE and plasmid sequencing results. In contrast, the fusion plasmid p17ZR-91-Vir-KPC was not detectable by S1-PFGE, suggesting that copy number of the fusion plasmid was much lower than that of two non-fusion plasmids in 17ZR-91. However, in transconjugant EC600-TC, the copy number of the fusion plasmid was similar to those of two non-fusion plasmids, enabling visualization of all three plasmids in S1-PFGE (Fig. 1a). To confirm the dynamic presence of fusion plasmid and non-fusion plasmids in transconjugants, we have screened around 50 transconjugants obtained from different conjugation experiments using 17ZR-91 as donor strain using PCR confirmation assays with results showing that the fusion and non-fusion plasmids were present in all these 50 transconjugants tested.

**Transmission of conjugative fusion plasmid in ST11 *K. pneumoniae*.** To test if the fusion plasmid p17ZR-91-Vir-KPC was able to be conjugated to other *K. pneumoniae* isolates, conjugation tests were carried out utilizing both 17ZR-91 and EC600-TC as donors and *K. pneumoniae* KP04-1 as recipient. Our data showed that p17ZR-91-Vir-KPC could be conjugated to clinical ST11 *K. pneumoniae* KP04-1, a classical *K. pneumoniae* strain. Transconjugants (KP04-1-TC) of this strain exhibited resistance to potassium tellurite and carbapenem and were found to carry the *rmpA2* and *bla*$_{KPC-2}$ gene (Table 1) with identical XbaI-PFGE profile as the recipient KP04-1 (Fig. 1a). S1-PFGE results illustrated that these transconjugants had obtained two extra plasmids with sizes same as two non-fusion plasmids in EC600-TC, but not the fusion plasmid p17ZR-91-Vir-KPC (Fig. 1a). To determine if p17ZR-91-Vir-KPC was present in these transconjugants, PCR assays targeting the fusion sites as described above were performed, with results showing that all four PCR products could be detected in these transconjugants (Supplementary Fig. S4b). This finding suggests that p17ZR-91-Vir-KPC was present in these transconjugants but may switch back and forth dynamically to two non-fusion plasmids. Similar to the case of 17ZR-91, the copy number of p17ZR-91-Vir-KPC is very low in these transconjugants, with only two non-fusion plasmids being detectable by S1-PFGE (Fig. 1a). In *E. coli*, the fusion plasmid and two non-fusion plasmids are present in similar proportion of copy number, whereas in *K. pneumoniae*, the copy number of fusion plasmid seems to be very low, which may be due to the adaption of this fusion plasmid in different bacterial species. Interestingly, we found that conjugation efficiency recorded when using EC600-TC as donor was much higher than using 17ZR-91 as donor. Such differences in efficacy might be due to the higher copy number of the fusion plasmid in EC600-TC than in 17ZR-91.

**Acquisition of p17ZR-91-Vir-KPC confers *K. pneumoniae* carbapenem-resistance and hypervirulence phenotype.** The MIC data indicated that upon acquisition of p17ZR-91-Vir-KPC, transconjugants became resistant to all β-lactams tested including carbapenems, indicating that genetic elements that encode resistance to these antibiotics had been acquired (Table 1). The virulence potential of strain 17ZR-91 and its transconjugants were evaluated in a mouse infection model. Our data showed that infection of mice with $5 \times 10^5$ CFU of strain 17ZR-91 can cause 40% mortality at 96 h, which was similar to the hypervirulent control strain HvKP1088 ($P = 0.610$) and significantly higher than HvKP4 ($P < 0.001$), but still much higher when compared with the mortality of low virulence control strain FJ8 (Fig. 1b). Infection of the mice with $5.0 \times 10^5$ CFU of transconjugant, KP04-1-TC, caused 60% mortality at 120 h, a level similar to that of HvKP4 ($P = 0.662$) but significantly lower than HvKP1088 ($P = 0.009$); while 0% mortality was recorded at 120 h when infection with same amount of KP04-1 or FJ8 (Fig. 1b). These results further verified that acquisition of virulence plasmid p17ZR-91-Vir-KPC by *K. pneumoniae* strain KP04-1 enabled it to evolve into a CR-HvKP strain. The data also suggested that the virulence level of ST11 CR-HvKP such as HvKP4 and KP04-1-TC, might be lower than K1 CR-HvKP strains such as HvKP1088 and 17ZR-91.

**Prevalence of conjugative plasmid encoding carbapenem resistance and hypervirulence in clinical CRKP strains.** We next investigated if carriage of the fusion plasmid p17ZR-91-Vir-KPC or co-carriage of plasmids p17ZR-91-Vir and p17ZR-91-KPC has become a common phenomenon in clinical *K. pneumoniae* strains by examining 200 clinical carbapenem-resistant

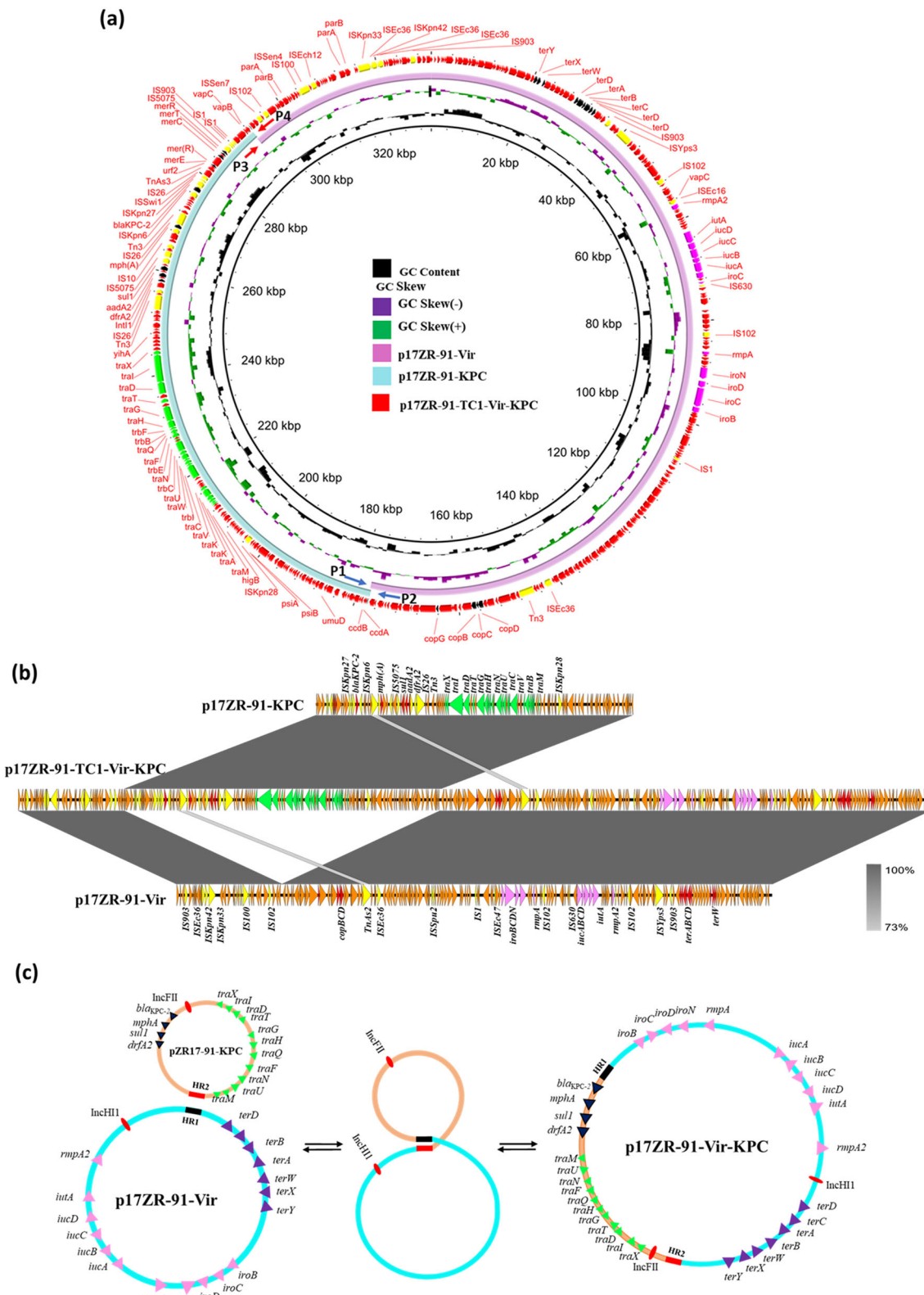

**Fig. 3 Formation of conjugative fusion virulence plasmid p17ZR-91-Vir-KPC.** Alignment of virulence plasmid p17ZR-91-Vir and $bla_{KPC-2}$-bearing plasmid p17ZR-91-KPC with fusion plasmid p17ZR-91-Vir-KPC recovered from *K. pneumoniae* 17ZR-91 using BRIG (**a**) and Easyfig (**b**). **c** Potential mechanism of plasmid fusion through homologous recombination. Two plasmids were integrated at the HR region (HR1, black and HR2, red) and underwent homologous recombination, resulting in a fusion plasmid.

*K. pneumoniae* strains collected from the hospital in which strain 17ZR-91 was recovered, we identified eight strains that contained p17ZR-91-KPC-like plasmids, in which the HR2 region was detectable (Supplementary Table S1, Supplementary Fig. S5). Among these eight strains, we found three of which also contained the virulence plasmid p17ZR-91-Vir, suggesting that these three strains have potential to transfer the p17ZR-91-Vir-KPC fusion plasmid to other *K. pneumoniae* strains (Supplementary Fig. S6). Two of these three strains, 17ZR-31 (isolated on March 02, 2017) and 17ZR-84 (isolated on April 16, 2017), belonged to ST86. Strain 17ZR-84 was found to share 23 SNPs with 17ZR-91, suggesting that these K2 CR-HvKP strains belonged to the same clone, and that members of this clone were circulating in the hospital environment. On the other hand, strain 17ZR-31 was found to belong to another distinct clone of K2 HvKP (Supplementary Table S2). Importantly, the third strain 17ZR-127 (isolated on May 12, 2017) was found to belong to ST11, a non-K1/K2 type (K64), suggesting that p17ZR-91-Vir and p17ZR-91-KPC could be transmitted to ST11 type of *K. pneumoniae* through formation of a conjugative fusion plasmid and generated non-K1/K2 ST11 HvKP, which become prevalent in clinical setting in China (Supplementary Table S1, Supplementary Fig. S6, S7a). Retrospective search of clinical record showed that strain 17ZR-127 was isolated from a blood sample of a 73-years-old male patient who was admitted into intensive care unit in 2017 with several underlying diseases including viral encephalitis, lung infection, respiratory failure, high blood pressure, and rectal cancer surgery. This patient has undergone tracheal intubation and tracheotomy procedure and been treated with cefoperazone/sulbactam and meropenem but with very poor outcome. The patient was discharged from the hospital with multiple organ failure and septic shock after 35 days of hospitalization. The virulence level of this strain was determined in mouse infection model with result showing that it exhibited lower virulence than ST11 CR-HvKP strain, HvKP4, but much higher virulence than ST11 CRKP strain FJ8 (Fig. 1b). Similar to 17ZR-91, conjugation experiments showed that the virulence and $bla_{KPC-2}$-bearing plasmids could be conjugated from clinical isolates, 17ZR-31, 17ZR-84, and 17ZR-127, to *E. coli* recipient EC600 with the efficiency of 1.96E-08, 1.75E-08, and 3.75E-08, respectively. To confirm that the fusion plasmid p17ZR-91-Vir-KPC was also present in these three clinical isolates, PCR assays as described above were performed in these three isolates, with results showing that all four PCR products were successfully detected, suggesting that p17ZR-91-Vir-KPC and two non-fusion plasmids were also present in these three clinical isolates. These data support that the fusion plasmid p17ZR-91-Vir-KPC could be readily transmitted from K2 CR-HvKP to non-K1/K2 *K. pneumoniae* strains in clinical settings.

In addition, we screened *K. pneumoniae* genome sequences deposited in GenBank for the presence of p17ZR-91-Vir and p17ZR-91-KPC, a total of 42 strains carrying p17ZR-91-KPC-like plasmids which also contain the homologous region (HR) were identified, among which *Klebsiella* spp constituted the majority; four *E. coli* and one *Enterobacter hormaechei* strains were also identified. Among these 42 strains, three also contained the HR region-bearing p17ZR-91-Vir plasmid. These three strains belonged to ST23 (K1), ST36 (K62), and ST86 (K2) types of *K. pneumoniae* and all were collected from different parts of China in 2015 and 2016 (Supplementary Table S1). The virulence plasmid harbored by these three strains were identical to p17ZR-91-Vir (Supplementary Fig. S6), whereas the other conjugative plasmid that they also harbored contained the $bla_{CTX-M}$ gene instead of $bla_{KPC-2}$. Except for the MDR region, the backbone of this plasmid was highly similar to p17ZR-91-KPC (Supplementary Fig. S7a). These three plasmids aligned well with a $bla_{CTX-M}$-

bearing plasmid, pR210-2-CTX (NZ_CP034085) (Supplementary Fig. S7b). Analysis of genomic data of these three strains suggested that the p17ZR-91-KPC-like plasmid might have been acquired by ST23 (K1) and ST86 (K2) HvKP strains, thereby facilitating transmission of both the virulence plasmid and p17ZR-91-KPC-like plasmid to other non-K1/K2 ST36 (K62) *K. pneumoniae* isolates and enabling such strains to evolve into MDR HvKP. Phylogenetic analysis of all 43 strains that carried p17ZR-91-KPC-like plasmid, including 17ZR-91, showed that these strains were genetically diverse, but ST15, ST11, and ST86 were more common. Most of these strains were isolated from clinical samples in China, with only a few from other countries (Supplementary Table S1, Fig. 4).

## Discussion

*K. pneumoniae* has become one of the most important clinical pathogens in recent years as it continued to evolve into multidrug resistant and hypervirulent genetic variants. Strains of specific genetic types such as K1/K2 serotypes first evolved into HvKP but remain susceptible to most antibiotics. Within the last decade, *K. pneumoniae* has also evolved to become multidrug resistant; in particular, strains that are resistant to the last line antibiotic carbapenem have emerged. The most common CRKP strains transmitted worldwide belongs to CG258, with ST258 being the most prevalent in Europe and US, whereas ST11 is the most common in Asian. In 2015, studies have reported the emergence of carbapenem resistant ST23 K1 HvKP strains[12,15,26,27], which have caused fatal infections even with aggressive antibiotic treatment. In 2017, our group reported the evolution of CRKP into CR-HvKP through acquisition of virulence plasmid from HvKP strains[18]. These newly emerged CR-HvKP strains express both carbapenem-resistance and hypervirulence phenotypes and cause rapidly progressed and invasive infections with extremely high mortality. These Hv-CRKP strains have now been reported in many countries with the highest prevalence in Asia[12,28–30]. Most importantly, these non-K1/K2 CR-HvKP strains, in particular the ST11 type of CR-HvKP, have quickly spread throughout China and severely undermined infection control effort. Recent studies reported the transmission of non-conjugative virulence plasmid through forming a conjugative fusion plasmid with a conjugative helper plasmid, while there is still lack of direct evidence of transmission of virulence from K1/K2 *K. pneumoniae* strains[31,32]. Our study revealed the process of transmission of the virulence plasmid from K2 HvKP to non-K1/K2 *K. pneumoniae*, generating non-K1/K2 CR-HvKP. In this process, K1/K2 HvKP strains might first obtain a conjugative $bla_{KPC-2}$-bearing plasmid, which will then fuse with a pLVPK-like virulence plasmid to generate a conjugative fusion plasmid. This fusion plasmid could be disseminated to non-K1/K2 *K. pneumoniae* to convert any *K. pneumoniae* strains with no existing antimicrobial resistance or virulence determinants to a carbapenem-resistant HvKP strain immediately.

Plasmid fusion has been reported in several studies, with majority of these known fusion processes being mediated by insertion sequence elements through homologous recombination[33–35]. The first reported hybrid resistance and virulence plasmid was obtained from a clinical HvKP strain collected in 2013 in China[15]. Plasmid pKP70-2 was found to harbor a $bla_{KPC-2}$-bearing fragment flanked by two copies of IS26 when compared to other pK2044-like virulence plasmids. Similarly, an instance of intermolecular transposition of $bla_{CTX-M-24}$ gene into the virulence plasmid mediated by a IS903D fragment was reported in China in 2019[36]. Lam et al. reported two hybrid virulence and resistance plasmids recovered from two clinical ST15 *K. pneumoniae* isolates in Norway[30]. Another hybrid

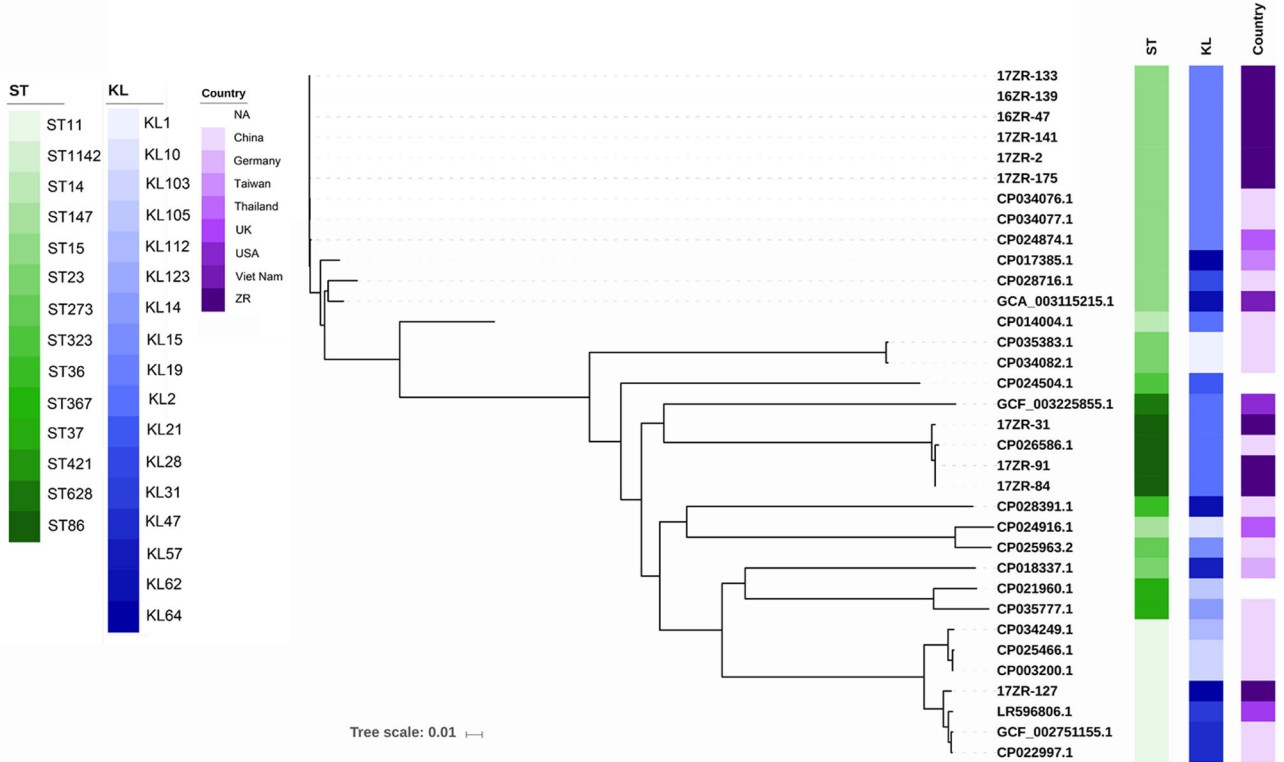

**Fig. 4 Phylogenetic analysis of bacterial strains carrying p17ZR-91-KPC like plasmids.** Strains with ZR in strain ID indicated that these CRKP strains were recovered from the same hospital in China as that of strain 17ZR-91; genomes of other strains, which were recovered from different countries, were retrieved from GenBank. Strains belonging to other genus such as *E. coli* were not included in phylogenetic analysis.

virulence plasmid pVir was recovered from an ST11 *K. pneumoniae* TVGHCRE225[29]. Plasmid pVir was a hybrid of a pPMK-NDM-like resistance plasmid and a pK2044-like virulence plasmid. Interestingly, although some virulence-associated genes such as *rmpA*, *rmpA2*, *iroBCDN* and *iucABCDiutA* existed in plasmid pVir, *K. pneumoniae* TVGHCRE225 did not express a high level of virulence in a mouse infection model. ColE1-type plasmid pIP843 recovered in 1996[37] contained an *oriT* locus, but genes encoding other conjugation functions were absent, resulting in the unsuccessful mobilization of this plasmid from the original *K. pneumoniae*. Interestingly, Lin et al. found a large conjugative plasmid pE66An from a clinical *E. coli* collected in the identical geographical region to pIP843[38]. Plasmid pE66An contained a genetic structure of a co-integrate of an original 73-kbp plasmid and pIP843, indicating that pE66An was generated upon co-integration of the original plasmid and pIP843[39]. This idea was consistent with a previous finding in which a plasmid pJHCMW1 contained just an *oriT* but could be mobilized by a co-resident plasmid[40]. However, it is expected that mobilization through this process occurs rarely in nature. A recent study also reported a similar phenomenon of dynamic presence of two small plasmids and one fusion plasmid in a *Salmonella* strain, where the fusion process was mediated by an IS*Pa40* through homologous recombination. One small plasmid encoded ciprofloxacin resistance and the other encoded ceftriaxone resistance. Fusion of these plasmids could form a conjugative plasmid encoding resistance to both ceftriaxone and ciprofloxacin, which are currently two major choices of treatment for *Salmonella* infections[25]. In this study, although only a 275 bp cognate fragment was detected in plasmid p17ZR-91-Vir and p17ZR-91-KPC, it could still mediate fusion of these two plasmids through homologous recombination to generate the fusion plasmid, enabling evolution

of the non-conjugative plasmid p17ZR-91-Vir into a conjugative plasmid. In the view of presence of various mobile elements in pLVPK-like virulence plasmid, it is likely that this plasmid has high potential to fuse with other conjugative MDR plasmids, and form conjugative MDR and virulence-encoding plasmids, facilitating transmission of virulence plasmid to different types of non-K1/K2 *K. pneumoniae* strains. In conclusion, our study revealed the emergence of a conjugative fusion plasmid that encoded both carbapenem resistance and hypervirulence phenotypes and deciphered an evolution mechanism of non-K1/K2 HvKP, prompting further monitoring of dissemination of CR-HvKP strains in clinical settings.

## Methods

**Bacterial isolation, species identification, and antimicrobial susceptibility tests.** *K. pneumoniae* 17ZR-91 characterized in this study was isolated from a sputum sample of a 74-years-old male patient in a hospital in Hangzhou, Zhejiang province, China in 2017. The patient was diagnosed with multiple organ failure including heart, lung, kidney and blood coagulation, and died in the hospital after treatment with various antibiotics including carbapenems. The strain was isolated 33 days after the hospitalization of the patient, suggesting the acquisition of this strain in hospital. Species identification was performed using the Vitek 2 system (bioMérieux, France) and MALDI-TOF MS apparatus (Bruker, Germany). Antimicrobial susceptibility tests were conducted by microdilution method recommended by Clinical and Laboratory Standards Institute[41]. *Escherichia coli* ATCC25922 was used as the quality control.

**Whole-genome sequencing and bioinformatics analysis.** Whole genome was obtained and sequenced by the NextSeq 500 Illumina platform (San Diego, CA) with 2 × 150 bp paired-end reads and Nanopore MinION (long-read) sequencing platform using the RBK004 sequencing kit and MinION R9.4.1 flow cell. Hybrid genome assembly with both short and long reads was conducted using Unicycler[42]. Sequence types were identified by Kleborate software according to their genetic variations in housekeeping determinants. Capsular type identification of assembled sequences were determined by Kaptive[43]. Virulence determinants were determined

by matching against BIGSdb *Klebsiella* genome database (http://bigsdb.pasteur.fr/klebsiella/klebsiella.html). Resistance genes and virulence loci were identified through the ResFinder[44] and mapping against BIGSdb *Klebsiella* genome database[45]. Sequence comparison of plasmids was conducted using BLAST Ring Image Generator (BRIG)[46] and Easyfig[47].

**Conjugation assay.** Conjugation tests were conducted to evaluate the transfer potential of virulence plasmid using rifampin-resistant *E. coli* EC600 and ciprofloxacin-resistant *K. pneumoniae* as the recipients. Briefly, *K. pneumoniae* 17ZR-91 and the recipient were cultured in fresh Luria Bertani (LB) broth at 37 °C for 4 h to reach the logarithmic phase. Then, the donor and recipient were mixed at a ratio of 1:4 and inoculated on a 0.45 μm membrane placed on a LB agar. After culturing for 24 h at 37 °C, bacterial mixture was incubated onto China Blue agar supplemented with 2 μg mL$^{-1}$ potassium tellurite ($K_2TeO_3$) and 600 μg mL$^{-1}$ rifampin when using *E. coli* EC600 as the recipient. In addition, conjugation experiment was conducted using EC600-TC as the donor strain and *K. pneumoniae* as recipient. Transconjugants were selected on MacConkey agar supplemented with 4 μg mL$^{-1}$ $K_2TeO_3$ and 2ug mL$^{-1}$ ciprofloxacin. Transconjugants were identified by PCR assay[27] and minimum inhibitory concentration (MIC) profiles. S1-PFGE and XbaI-PFGE were conducted as previously reported[48] to verify the acquisition of plasmid by corresponding recipient. Conjugation efficiency was determined as the ratio of the number of transconjugants and recipients.

**PCR assay.** PCR tests were conducted to detect the existence of the fusion plasmid and two non-fusion plasmids as described previously[49]. The primers used were as follows: 1F- GACGAATGGCGCGGTTATTG; 1R-TGATGAATCACGCAGCCACA; 2F-GCGGAAAACAGAACCGTGAC; and 2R-CTGATTTTCAGCCTGCCAGC. Amplification was carried out as follows: initial denaturation at 95 °C for 5 min, 35 cycles of 95 °C for 30 s, 55 °C for 30 s and 72 °C for 1 min; and a final elongation step at 72 °C for 10 min. PCR tests targeting the fusion regions were described in Figure S4a. Briefly, PCR1 (1 F/1 R) and PCR2 (2 F/2 R) were designed to confirm the presence of fusion plasmid p17ZR-91-Vir-KPC with product sizes of 716 bp and 759 bp respectively. PCR3 (1 F/2 R) was designed to detect the presence of p17ZR-91-Vir with product size of 783 bp; and PCR4 (2 F/1 R) was designed to detect the presence of p17ZR-91-KPC with product size of 692 bp. Being positive in all the four PCR tests indicates the existence of fusion plasmid and two non-fusion plasmids in the same strain.

**Mouse infection assay.** A mouse infection model was applied to evaluate the virulence discrepancy of *K. pneumoniae* isolates. Male NIH mice (4 weeks-old, 15 g approximately) from Guangdong Center for Experimental Animals (Guangzhou, China) were used in this study (NIH mice, bred by the National Institutes of Health, are outbred albino mice, which are widely used in pharmacology, toxicology, and bacteriology research.). Ten mice ($n = 10$) randomly allocated into every experimental group were infected with $5.0 \times 10^5$ CFU of each *K. pneumoniae* isolate by intravenous injection. Survival of mice in each group was recorded for 120 h post infection. An ST11 CR-HvKP isolate HvKP4 and an ST23 K1 HvKP isolate HvKP1088 that were validated as HvKP strains in previous studies were included as hypervirulence control and an ST11 CRKP isolate FJ8 that did not harbor a virulence plasmid was also included as low virulence control[27]. Animal infection tests were conducted at least twice to confirm the data consistency. Survival curves were depicted by GraphPad Prism 7 software with statistical analysis performing by the Log-rank (Mantel-Cox) test.

**Statistics and reproducibility.** Statistical analysis was conducted with GraphPad Prism 7. The data of conjugation experiments were obtained from at least three independent experiments. Animal infection tests were conducted at least twice to confirm the consistency of the data. In animal infection tests, survival curves were analyzed by the Log-rank (Mantel-Cox) test.

**Ethics statement.** Animal infection tests were approved by Animal Subjects Ethics Sub-Committee of The Hong Kong Polytechnic University.

**Reporting summary.** Further information on research design is available in the Nature Research Reporting Summary linked to this article.

## Data availability
Complete sequences of plasmids p17ZR-91-Vir, p17ZR-91-KPC, and p17ZR-91-Vir-KPC have been deposited in the GenBank databases under accession numbers MN200128, MN200129 and MN200130, respectively. All other data associated with this study are available upon request.

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

## Acknowledgements

This study was funded by Collaborative Research Fund from the Research Grant Council of the Government of Hong Kong SAR (C5026-16G), Research Impact Fund (R5011-18F) and National Natural Science Foundation of China (81772250 and 81601815).

## Author contributions

MMX designed and performed the experiments; X.M.Y. and Q.X. participated in con-jugation experiments; K.C.C. and Z.W.Z. helped with animal experiments, L.W.Y. and N.D. helped with genome sequencing; Q.L.S., L.B.S., D.X.G. helped with collection of clinical strains and data; X.M.Y. and R.Z. helped with study design and GenBank data analysis; E.W.C.C. edited the manuscript; S.C. supervised the whole project and wrote the manuscript.

## Competing interests

The authors declare no competing interests.
