## [Peer Review File · Communications Biology]

Reviewers' comments:

Reviewer #1 (Remarks to the Author):

This manuscript describes a mechanism for the development of a conjugal hybrid/fusion plasmid in a ST86 hypervirulent *Klebsiella pneumoniae* (hvKp) isolate that contains both antimicrobial resistance genes and hvKp-specific virulence genes. Data is also presented that supports this hybrid plasmid was transferred to an ST11 isolate. These data identify a mechanism by which classical *K. pneumoniae* (cKp) strains could acquire genes from the hvKp-specific virulence plasmid and thereby increase their virulence potential. Obviously if this occurs in an extensively drug resistant cKp strain this would be problematic. New and important data is reported, which supports the conclusions by the authors. I have but a few minor comments.

1. line 52. "capsular" should be "capsule". line 237. "donor" should be "donors". There are several other word usage and grammatical issues that need to be addressed.

2. lines 91-95. Additional clinical details if available should be included. What was the infectious syndrome? Were there multiple sites of infection? Was the infection community acquired?

3. lines 285-288. Are there any clinical details available for 17ZR-127? It would be interesting if infection caused by this strain had features of infection seen when the infecting strain has the hvKp phenotype (e.g. multiple sites of infection, metastatic spread, unusual sites for cKp et al).

4. line 136. Details on the "NIH mice" should be given such as inbred or outbred, genotypic and phenotypic features (if any) et al.

Line 139-140. Please give some details on HvKp4, HvKp1088, and KP04-1

5. line 231 and throughout. The term "small plasmids" is used. p17ZR-91-Vir is not small. "Non-fusion" or "non-hybrid" or a suitable alternative should be used.

line 242. Likewise, the term "large plasmid" is used. Perhaps "fusion" or "hybrid" or a suitable alternative should be used.

6. line 253. "differences in efficacy" would be clearer than "discrepancy".

7. lines 265-268. The result that KP04-1-TC and hvKp4 were less virulent compared to HvKP1088 and 17ZR-91 is interesting, but perhaps not surprising. There should be some discussion on this observation.

8. It would be interesting and important to assess 17ZR-127 in the murine infection model.

9. line 295. "These data confirm" is a bit overstated since transmission from K2 CR-HvKP is inferential. Perhaps "These data support" would be more appropriate.

Reviewer #2 (Remarks to the Author):

Manuscript entitled "Clinical evolution of ST11 carbapenem resistant and hypervirulent *Klebsiella pneumoniae*" by Xie et al. gave a very explanatory and interesting analysis of a newly emerged carbapenem resistant hypervirulent *K. pneumoniae* (CR-HvKP) strain that expresses both carbapenem resistance and hypervirulence phenotypes and could cause serious invasive infections with high mortality rates.

This fusion of conjugative plasmids and the conjugation experiments are presented nicely in this manuscript and again it warns against the worsening of antibiotic resistance crisis.

Also in the materials and methods, authors went to PCR, without mentioning the gene of question or the expected product size, so this needs clarification.

I may suggest more clarifications to the figures like mentioning the source of the TC of each

isolate. Also the S1-PFGE figure may also need more illustration in the findings like commenting on the number of plasmids in each isolate.

The explanation and the analysis are relevant and the language is perfect.

I do recommend accepting the article as it addresses a very hot research point and warns the progression of MDR in the region and very likely worldwide.

Reviewer #3 (Remarks to the Author):

The authors present a very nice study of potentially very significant clinical importance showing co-conjugation of hypervirulence and MDR plasmids of *Klebsiella*. Most significantly they show that this has occurred in the hospital setting in a clinical ST11 strain.

I have some comments I would encourage the authors to address:

Major Points:

Line 130: PCR assays for what exactly?

Line 238: I feel it is very important to capture an idea of how often this fusion plasmid arises during conjugations. If this is to be considered an important clinical observation we need to know how often the fusion is formed and becomes a phenotype bearing entity during conjugation. I would suggest around 50 conjugations performed and the PCR assay used to determine how often the fusion forms, and then from that how often it is conferring both hypervirulent and MDR phenotypes

Line 250: Could it be that the hypervirulent plasmid is not fully recombining as such but rather there is conjugation hitchhiking going on as happens for example in HPis? This needs to be included as a possibility. This could also be tested by isolating the fusion plasmid and then assaying how stable it is, or how often it disassembles in a large number of repeat conjugations.

Line 382: I am nervous of this sentence as there is no evidence or data showing fusion with a large number of MDR plasmids.

Minor Points:

Line 48: Change "reason" to "cause"

Line 53: change "capsular" to "capsule"

Line 60: Change beginning of sentence to "Additionally, certain lineages...."

Line 62: Change to "and is regarded"

Line 63: Remove "For many years" from start of sentence

Line 66: change beginning of sentence to "Recent studies have..."

Line 71: Remove "In another evolution path" from beginning of sentence

Line 72: change to "have been shown to acquire the virulence plasmid....."

Line 77: change to "enormous challenges"

Line 81: change to "the virulence plasmid"

Line 82: change to "conjugation. This has probably limited the transmission of....."

Line 85: change to "encodes"

Line 85: change to "hypervirulence, uncovering a mechanism of evolution towards HvKP in non-K1/K2 strains"

Line 87: change "would" to "could"

Line 94: change to "diagnosed with"

Reviewers' comments:

Reviewer #1 (Remarks to the Author):

This manuscript describes a mechanism for the development of a conjugal hybrid/fusion plasmid in a ST86 hypervirulent *Klebsiella pneumoniae* (hvKp) isolate that contains both antimicrobial resistance genes and hvKp-specific virulence genes. Data is also presented that supports this hybrid plasmid was transferred to an ST11 isolate. These data identify a mechanism by classical *K. pneumoniae* (cKp) strains could acquire genes from the hvKp-specific virulence plasmid and thereby increase their virulence potential. Obviously if this occurs in an extensively drug resistant cKp strain this would be problematic. New and important data is reported, which supports the conclusions by the authors. I have but a few minor comments.

1. line 52. “capsular” should be “capsule”. line 237. “donor” should be “donors”. There are several other word usage and grammatical issues that need to be addressed.

Response: Line 52. “capsular” changed to “capsule”.

Line 237. “donor” changed to “donors”.

Several other word usage and grammatical issues were also revised.

2. lines 91-95. Additional clinical details if available should be included. What was the infectious syndrome? Were there multiple sites of infection? Was the infection community acquired?

Response: The clinical data have been added in the manuscript.

3. lines 285-288. Are there any clinical details available for 17ZR-127? It would be interesting if infection caused by this strain had features of infection seen when the infecting strain has the hvKp phenotype (e.g. multiple sites of infection, metastatic spread, unusual sites for cKp et al).

Response: Some clinical details have been added. The sentences “Retrospective search of clinical record showed that strain 17ZR-127 was isolated from a blood sample of a 73-years-old male patient who was admitted into intensive care unit in 2017 with several underlying diseases including viral encephalitis, lung infection, respiratory failure, high blood pressure and rectal cancer surgery. This patient has undergone tracheal intubation and tracheotomy procedure and been treated with cefoperazone / sulbactam and meropenem but with very poor outcome. The patient was discharged from the hospital with multiple organ failure and septic shock after 35 days’ hospitalization.” were added.

4. line 136. Details on the “NIH mice” should be given such as inbred or outbred, genotypic and phenotypic features (if any) et al.

Response: The sentence “NIH mice, bred by the National Institutes of Health, are outbred albino mice, which are widely used in pharmacology, toxicology and bacteriology research.” was added.

5. Line 139-140. Please give some details on HvKp4, HvKp1088, and KP04-1

Response: revised accordingly. Strain KP04-1 was a randomly selected carbapenem-sensitive K. pneumoniae that did not harbor a virulence plasmid.

6. line 231 and throughout. The term “small plasmids” is used. p17ZR-91-Vir is not small. “Non-fusion” or “non-hybrid” or a suitable alternative should be used.

Response: revised accordingly. The term “small plasmids” are revised to “non-fusion plasmids” throughout the manuscript.

7. line 242. Likewise, the term “large plasmid” is used. Perhaps “fusion” or “hybrid” or a suitable alternative should be used.

Response: Line 242. “large” changed to “fusion”.

8. line 253. “differences in efficacy” would be clearer than “discrepancy”.

Response: Line 253. “discrepancy” changed to “differences in efficacy”.

9. lines 265-268. The result that KP04-1-TC and hvKp4 were less virulent compared to HvKP1088 and 17ZR-91 is interesting, but perhaps not surprising. There should be some discussion on this observation.

Response: This has been discussed in the text.

10. It would be interesting and important to assess 17ZR-127 in the murine infection model.

Response: The data has been added.

11. line 295. “These data confirm” is a bit overstated since transmission from K2 CR-HvKP is inferential. Perhaps “These data support” would be more appropriate.

Response: Line 295. “confirm” changed to “support”.

Reviewer #2 (Remarks to the Author):

Manuscript entitled “Clinical evolution of ST11 carbapenem resistant and hypervirulent *Klebsiella pneumoniae*” by Xie et al. gave a very explanatory and interesting analysis of a newly emerged carbapenem resistant hypervirulent *K. pneumoniae* (CR-HvKP) strain that expresses both carbapenem resistance and hypervirulence phenotypes and could cause serious invasive infections with high mortality rates.

This fusion of conjugative plasmids and the conjugation experiments are presented nicely in this manuscript and again it warns against the worsening of antibiotic resistance crisis.

1. Also in the materials and methods, authors went to PCR, without mentioning the gene of question or the expected product size, so this needs clarification.

Response: revised accordingly.

The first sentence “PCR assays were performed as described previously.” changed to “PCR assays were performed to detect the presence of the fusion plasmid and two non-fusion plasmids as described previously.”.

The sentences “PCR tests targeting the fusion regions were described in Figure S4a. Briefly, PCR1 (1F/1R) and PCR2 (2F/2R) were designed to confirm the presence of fusion plasmid p17ZR-91-Vir-KPC with product sizes of 716bp and 759bp respectively. PCR3 (1F/2R) was designed to detect the presence of p17ZR-91-Vir with product size of 783bp; and PCR4 (2F/1R) was designed to detect the presence of p17ZR-91-KPC with product size of 692bp. Being positive in all the four PCR tests indicates the presence of fusion plasmid and two non-fusion plasmids in the same strain.” were added.

2. I may suggest more clarifications to the figures like mentioning the source of the TC of each isolate. Also, the S1-PFGE figure may also need more illustration in the findings like commenting on the number of plasmids in each isolate.

Response: The source of the EC600-TC and KP04-1-TC and the numbers of plasmids in these two strains have been added both in the text and figure legend.

3. The explanation and the analysis are relevant and the language is perfect.

I do recommend accepting the article as it addresses a very hot research point and warns the progression of MDR in the region and very likely worldwide.

Response: Thanks for the comments.

Reviewer #3 (Remarks to the Author):

The authors present a very nice study of potentially very significant clinical

importance showing co-conjugation of hypervirulence and MDR plasmids of Klebsiella. Most significantly they show that this has occurred in the hospital setting in a clinical ST11 strain.

I have some comments I would encourage the authors to address:

Major Points:

1. Line 130: PCR assays for what exactly?

Response: revised accordingly.

The first sentence "PCR assays were performed as described previously." changed to "PCR assays were performed to detect the presence of the fusion plasmid and two non-fusion plasmids as described previously."

The sentences "PCR tests targeting the fusion regions were described in Figure S4a. Briefly, PCR1 (1F/1R) and PCR2 (2F/2R) were designed to confirm the presence of fusion plasmid p17ZR-91-Vir-KPC with product sizes of 716bp and 759bp respectively. PCR3 (1F/2R) was designed to detect the presence of p17ZR-91-Vir with product size of 783bp; and PCR4 (2F/1R) was designed to detect the presence of p17ZR-91-KPC with product size of 692bp. Being positive in all the four PCR tests indicates the presence of fusion plasmid and two non-fusion plasmids in the same strain." were added.

2. Line 238: I feel it is very important to capture an idea of how often this fusion plasmid arises during conjugations. If this is to be considered an important clinical observation, we need to know how often the fusion is formed and becomes a phenotype bearing entity during conjugation. I would suggest around 50 conjugations performed and the PCR assay used to determine how often the fusion forms, and then from that how often it is conferring both hypervirulent and MDR phenotypes

Response: Thanks for your great comment. From our PCR assays targeting to the fusion plasmid, we showed that the fusion plasmid is present together with two non-fusion plasmids in a dynamic form in clinical K. pneumoniae strains with the fusion plasmid being in very low copy number; therefore it could not be detected by S1-PFGE. Interestingly, in E. coli transconjugants, the fusion plasmid and two non-fusion plasmids are also present in dynamic form, while the copy number of the fusion plasmid is as high as two non-fusion plasmids, therefore we can see both the fusion plasmids and two non-fusion plasmids by S1-PFGE. In conclusion, the fusion plasmid is stably present in both K. pneumoniae and E. coli at all time.

3. Line 250: Could it be that the hypervirulent plasmid is not fully recombining as such but rather there is conjugation hitchhiking going on as happens for example in HPIs? This needs to be included as a possibility. This could also be tested by isolating the fusion plasmid and then assaying how stable it is, or how often it disassembles in a large number of repeat conjugations.

Response: Our response for your previous question could address this comment. The fusion plasmid is present all the time in clinical K. pneumoniae strains.

4. Line 382: I am nervous of this sentence as there is no evidence or data showing fusion with a large number of MDR plasmids.

Response: Thanks for the comment. We have toned it down by deleting a large number of MDR plasmid and changed it to other conjugative MDR plasmids.

Minor Points:

5. Line 48: Change "reason" to "cause"

Response: revised accordingly.

6. Line 53: change "capsular" to "capsule"

Response: revised accordingly.

7. Line 60: Change beginning of sentence to "Additionally, certain lineages...."

Response: revised accordingly.

8. Line 62: Change to "and is regarded"

Response: revised accordingly.

9. Line 63: Remove "For many years" from start of sentence

Response: revised accordingly.

10. Line 66: change beginning of sentence to "Recent studies have..."

Response: revised accordingly.

11. Line 71: Remove "In another evolution path" from beginning of sentence

Response: revised accordingly.

12. Line 72: change to "have been shown to acquire the virulence plasmid....."

Response: revised accordingly.

13. Line 77: change to "enormous challenges"

Response: revised accordingly.

14. Line 81: change to "the virulence plasmid"

Response: revised accordingly.

15. Line 82: change to "conjugation. This has probably limited the transmission of...."

Response: revised accordingly.

16. Line 85: change to "encodes"

Response: revised accordingly.

17. Line 85: change to "hypervirulence, uncovering a mechanism of evolution towards HvKP in non-K1/K2 strains"

Response: revised accordingly.

18. Line 87: change "would" to "could"

Response: revised accordingly.

19. Line 94: change to "diagnosed with"

Response: revised accordingly.

REVIEWERS' COMMENTS:

Reviewer #1 (Remarks to the Author):

Previous minor concerns have been adequately addressed.

Reviewer #3 (Remarks to the Author):

In the interests of fairness I have reviewed this revised manuscript purely in light of my own comments on the first submission and not those of the other reviewers.

I thank the authors for their clarification on my 2nd and 3rd points however it is not clear to me that these points have been incorporated in edits to the manuscript text. I raised the original comments as it was not clear to me in the manuscript they were being addressed and so i would strongly recommend weaving the response to my second comment into the discussion such that other readers will not be left with the same query.

A simple statement as provided in the response around observations in *K. pneumoniae* and *E. coli* would suffice, and clarity on how many times the conjugations were performed and the observation made

Reviewers' comments:

Reviewer #3 (Remarks to the Author):

In the interests of fairness I have reviewed this revised manuscript purely in light of my own comments on the first submission and not those of the other reviewers.

I thank the authors for their clarification on my 2nd and 3rd points however it is not clear to me that these points have been incorporated in edits to the manuscript text. I raised the original comments as it was not clear to me in the manuscript they were being addressed and so i would strongly recommend weaving the response to my second comment into the discussion such that other readers will not be left with the same query.

A simple statement as provided in the response around observations in *K. pneumoniae* and *E. coli* would suffice, and clarity on how many times the conjugations were performed and the observation made.

Response: We have included this sentence in the revised manuscript "To confirm the dynamic presence of fusion plasmid and non-fusion plasmids in transconjugants, we have screened around 50 transconjugants obtained from different conjugation experiment using 17ZR-91 as donor strain using PCR confirmation assay with results showing that the fusion and non-fusion plasmids were present in all these 50 transconjugants tested."